# Initial Validation of the Ambivalent Sexism Inventory in a Military Setting

Vesna Trut [1,*], Petra Sinovčić [2,3] and Boris Milavić [3]

1    Croatian Defence Academy "Dr. Franjo Tuđman", 10000 Zagreb, Croatia
2    Organization for Promoting of Women's Rights "Domine", 21000 Split, Croatia; petsin@kifst.hr
3    Faculty of Kinesiology, University of Split, 21000 Split, Croatia; boris.milavic@kifst.hr
*    Correspondence: vesna.m.trut@gmail.com

**Abstract:** The military is a work environment in which the underrepresentation of women and the presence of gender prejudices continue to exist. The present study validated the *Ambivalent Sexism Inventory (ASI)* for the Croatian military population. To determine the ASI scale's basic metric characteristics, data were collected from a sample of 895 active-duty military personnel (445 men and 450 women). The study results determined satisfactory measurement characteristics for the ASI scale and confirmed the *ambivalent sexism* construct with its basic dimensions. Sexism in the military environment was found to a moderately high degree, and gender differences were observed. Three types of sexism endorsement were identified in both subsamples (*egalitarian*, *moderate egalitarian* and *traditional* for women, and *moderate egalitarian*, *traditional* and *hostile* for men), with additional differences detected in their socio–demographic and professional characteristics. The findings support the apparent exposure of women to sexism in the military environment, and suggest the need to raise awareness of the negative impact of gender prejudice on gender relations in the military.

**Keywords:** benevolent sexism; gender equality; gender relations; hostile sexism; masculine stereotypes; women in military

## 1. Introduction

Sexism is most often defined as "individuals' attitudes, beliefs, and behaviors, and organizational, institutional, and cultural practices that either reflect negative evaluations of individuals based on their gender or support unequal status of women and men" (Swim and Hyers 2009, p. 407). Numerous research studies related to this problem have been conducted thus far, and numerous laws and policies have been implemented with the aim of deterring incidents of sexism in society. Nevertheless, the problem continues to exist, and it remains considerably pervasive (Becker et al. 2014; Dick 2013; Pacilli et al. 2019). Several studies have suggested that gender roles, gender stereotypes and sexism are essential for women's educational and professional choices and opportunities (Pološki Vokić et al. 2019; Ridgeway and Correll 2004; Ridgeway 2009). Sexism in the workplace has a negative impact on work performance and employees' wellbeing (Barreto and Ellemers 2005, 2013; Heilman 2012; Heilman et al. 2004; Manuel et al. 2017), and it can be handled effectively with interventions aimed at improving gender equality in the workplace (Sojo et al. 2015). The armed forces have always been a male-dominated profession, and despite the positive changes in recent times, unequal gender distribution is still one of its major features (NATO 2019). However, equality between men and women has still been increasingly indicated as a significant precondition for the development of modern armed forces in recent times (Arostegui 2015; Quesada 2011). Accordingly, reforms in the security sector have occurred that resulted in changes within the military profession, including transformations in the nature of military tasks (Carreiras 2015). These tasks are no longer predicated exclusively upon physical strength, while modernization and

technological advancements have prompted the creation of military work positions that are now neither combatant nor gender-specific. As a consequence, traditional stereotypes about the "warrior soldier" have started to change, which has increased women's interest in military service. Still, women continue to represent a considerably smaller proportion of the total active-duty personnel, it continues to be difficult for them to have access to many military duties, and they are still less likely to be promoted to senior command and leadership positions (Carreiras 2015; Heinecken 2017; Kümmel 2015; NATO 2019).

### 1.1. Theoretical Background

Glick and Fiske (1996) developed a multidimensional construct of ambivalent sexism to redefine the previous definition of sexism, which was limited to negative attitudes and hostility towards women (Ramiro-Sánchez et al. 2018). The authors argued that prejudices against women included two interrelated and complementary ideologies that served to perpetuate gender inequalities that are found in different societies. They distinguished between hostile sexism (HS)—in other words, attitudes about women's inferiority to men, with stereotypical and negative views of women—and benevolent sexism (BS), or subjectively positive attitudes about the need to protect and idealize women and praise their "natural" role as mothers and romantic partners. Although intuitively positive, BS also reflects beliefs about the inferior role of women in relation to men. Glick and Fiske (1996, 2011) assumed that sexism is predicated upon social and biological differences between men and women. It stems from their inevitable interdependence, which is most pronounced in paternalism, gender differentiation, and heterosexual relationships. Hostility or benevolence occurs depending on whether competition or cooperation prevails in the relationship. The power a man exerts can be manifested as domination or protective paternalism. Differences in the characteristics and roles of women can be attributed to rivalry or complementarity, whereas heterosexual relationships can be perceived as rivalry or as intimate (Mikołajczak and Pietrzak 2015).

A cross-cultural study conducted on 19 national samples confirmed (1) the positive correlation between BS and HS, (2) gender differences in approving of sexism, and (3) the association of sexism with measures of gender inequality (Glick et al. 2000). Brandt (2011) analyzed longitudinal data collected in 57 countries and found that sexism not only legitimizes the societal *status quo*, but also directly intensifies a discriminatory gender hierarchy. Glick and Fiske (1996, 2001, 2011) believed that sexism affects interpersonal relationships among men and women in a variety of contexts, including the work environment. They argued that in work environments, where male dominance is "threatened" by gender equality, sexism is directed toward women who have taken traditional male roles. Furthermore, the military is also a workplace with stereotypical beliefs about the traits of women and men (Heinecken 2017; Karim and Beardsley 2013; Peters et al. 2015; Sakallı Uğurlu and Özdemir 2017; Weitz 2015).

### 1.2. Croatian Military Setting

The Croatian Armed Forces (CAF) emerged in the 1990s in the midst of the breakup of socialist Yugoslavia and the war for Croatian independence, during which the transition from socialism to democracy also started. The democratization process affected the military and caused major changes, which led to its professionalization. In 2009, Croatia joined NATO, and in 2013, the European Union. Simultaneously, the issue of gender equality in the military emerged (Franc et al. 2010; Trut and Milavić 2013). Although a greater proportion of women joined the CAF during the Homeland War, and the increasing trend in the number of women continued after that, women currently make up only about 13% of the total active-duty personnel. Similarly to other NATO armed forces, there is evidence of horizontal and vertical gender segregation in the CAF (Trut 2021; NATO 2019).

Studies of ambivalent sexism towards women in the military population are relatively rare (Ivarsson et al. 2005; Young and Nauta 2013; Sakallı Uğurlu and Özdemir 2017). Therefore, we wanted to conduct a study to investigate the presence of ambivalent sexism

in the CAF. Specifically, the present study intended to (1) validate the *Ambivalent Sexism Inventory* (ASI) for the Croatian military population; (2) examine gender differences in the approval of HS and BS; (3) identify individual characteristics of the service members that may be related to sexism; and (4) determine practical implications of the study results in a military work environment. Comparable to previous research, we assumed that HS would be endorsed by military men to a greater extent than by military women (Glick and Fiske 1996; Glick et al. 2000; Fernández et al. 2004; Brandt 2011). Furthermore, we hypothesized that military women would endorse BS to greater extent than military men due to their increased need for protection, idealization and affection (Glick et al. 2000; Moya et al. 2007). Additionally, we hypothesized that sexism could be predicted by socio–demographic, professional, and religious variables, which is in line with the prior research findings (Brajdić-Vuković et al. 2007; Galić and Nikodem 2006; Ivarsson et al. 2005; Marinović Jerolimov and Ančić 2014; Vučković Juroš et al. 2014; Zakrisson et al. 2012). Finally, we expected that these study results will contribute to a better understanding of gender relations in the military and to the improved implementation of *gender equality policy* in the CAF.

## 2. Materials and Methods

### 2.1. Participants

The study was conducted among members of the active-duty personnel of the CAF. Since women in 2019 constituted merely about 12% of the total service members, this study used a disproportionate stratified sample and a random sampling procedure. The stratification variables were *gender* and *professional status* in the military. The participants were selected from a list of men and a list of women that were sorted alphabetically and by professional status, from the highest to the lowest rank. The proportion of women and men within each stratum was determined based on their percentage in each of the professional categories in the military. The systematic random sample interval was determined separately within each stratum. Thus, approximately equal sample sizes for men and women were guaranteed, and statistical prerequisites for group comparison were met. The initial sample comprised 934 participants who completed the questionnaire. However, 39 participants were later excluded from the analyses due to missing values for more than 5% of the questions. The final sample comprised 445 men and 450 women. Categories of military personnel and rank insignia in the CAF were prescribed by national legislation, and for ease of understanding and comparison, appropriate NATO standardized insignia (NATO 2021) were listed. The male subsample predominantly involved soldiers (39.3%) [NATO ranks OR-1 to OR-3], followed by NCOs (38.9%) [NATO ranks OR-4 to OR-9], junior officers (12.1%) [NATO ranks OF-1 to OF-2] and the least senior officers (9.7%) [NATO ranks OF-3 to OF-9]. The female subsample comprised roughly equal numbers of soldiers (32.1%) [NATO ranks OR-1 to OR-3] and NCOs (32.6%) [NATO ranks OR-4 to OR-9], followed by junior officers (23.2%) [NATO ranks OF-1 to OF-2] and senior officers (12.1%) [NATO ranks OF-3 to OF-9]. Significantly more men than women held command/management positions (32.6% versus 21.7%). The average age of the study participants was 37.8 (ranging from 20 to 60 years). With respect to education, 65.1% of participants completed *high school* (74.1% of men vs. 56.2% of women), and 34.9% of them completed a *bachelor's degree* or higher (25.8% of men vs. 43.8% of women). The majority of participants lived in a *matrimonial and/or partnership union* (64.3%), while there were comparatively more women (38.0%) than men (33.4%) among single participants. The majority of participants *did not have children* (44.2%), and 9.1% had *three or more children*. Most participants grew up in *rural areas* (41.4%) and in *smaller towns* with up to 100,000 inhabitants (38.8%). Finally, 86.7% of all participants reported that they were *religious*, 11.5% *nonreligious*, and only 1.8% *convinced atheists*.

### 2.2. Measures

*Ambivalent Sexism Inventory, ASI* (Glick and Fiske 1996): The Croatian version of ASI (Tomić et al. 2013) was used to measure sexism towards women. The original scale consisted

of 22 *items* and had two subscales with 11 statements in each—one measuring HS and the other measuring BS. The scales used a 6-point Likert scale with the degree of agreement ranging from 0 (*strongly disagree*) to 5 (*strongly agree*). For the most adequate translation into Croatian, the content of the statement was modified and phrased as appropriate. Therefore, four of the six items, which were expressed as negative claims in the original questionnaire, were used as negative items in the Croatian translation of the questionnaire. For the remaining two originally negative items, the assessment direction was reversed in the translation, and they were reworded as positive items. After the inversion of all negatively formulated items, the result was expressed as an item's average result, with a higher result indicating a higher level of sexism. Glick and Fiske (1996) cited the possibility of using the results as two separate measures (HS and BS), as well as one integral ASI measure. Numerous studies have found satisfactory reliability of both scales (Glick and Fiske 1996; Glick et al. 2000; Fiske and North 2014), with *Cronbach's Alpha* ranging from 0.80 to 0.92 for the HS scale, and 0.73 to 0.85 for the BS scale.

*Religiosity:* Religiosity was measured with 2 items taken from the *European Values Study*, relating to two aspects of religiosity—*religious practice* and *faith importance* (Baloban et al. 2014). One item was used as an indicator of *religious practice*, in which the frequency of participating in religious rituals was estimated on a scale ranging from 1 (*almost never*) to 7 (*more than weekly*). One item was used as an indicator of *faith importance*, in which the participants assessed the relevance of faith on a scale ranging from 1 (*completely irrelevant*) to 10 (*very important*). The results were used as two separate measures of religiosity, in which a higher score reflected a higher level of religiosity.

*Socio–demographic and professional characteristics:* Common socio–demographic questions included measures that were assumed to be related to sexist prejudices among military personnel. The participants were asked about their *gender*, *age*, *educational level*, *marital status*, *number of children*, and *the age of the youngest child*. The *educational level* was evaluated as: 1—completed *high school*; 2—completed *bachelor's degree* or higher. *The place of growing up* was rated by participants on a scale of 1 to 5 (1—v*illage*; 2—*town up to 10,000 inhabitants*, 3—*town up to 100,000 inhabitants*; 4—*town up to 500,000 inhabitants*; 5—*town over 500,000 inhabitants*). *Professional status* in the military was evaluated by answers ranging from 1 to 4 (1—*soldier*, 2—*non-commissioned officer*, 3—*junior commissioned officer*, 4—*senior commissioned officer*). Participants were also asked about their current *command or leading duties* and about their years of their *military service*.

### 2.3. Procedures

This study is part of broader research aimed at investigating the determinants of gender equality in the CAF (Trut 2021). Its implementation was approved by the *Ethics Committee* of the Faculty of Law, University of Zagreb. Subsequently, the Ministry of Defense of the Republic of Croatia gave its approval to conduct the research among military personnel. The sampling procedure was completed in cooperation with the Military Administrative Department to select data from the Personnel Management Database. In the subsequent step, the chain of command enabled the conducting of research in all CAF units. Data were collected from February to June 2019 in all units and entities of the CAF, according to a previously prepared list of randomly selected active-duty personnel. All potential participants were informed in advance about the time and place of the survey. The survey was conducted by military psychologists who provided adequate space in each unit for group testing. The survey was anonymous, and it was administered in smaller groups of 15 participants on average. The participants were informed about the purpose and content of the study, how their data would be handled, and whether there were any potential risks associated with it. All participants gave their informed consent for inclusion prior to their participation in the study.

*2.4. Data Analysis*

Since the aim of this research was to validate the ASI in a military setting, the basic psychometric characteristics were determined. To determine both the latent structure and the *homogeneity* of the scales, exploratory *Principal Component Analysis* (PCA) with orthogonal *Varimax* rotation was used, applying the *Guttmann–Kaiser* criterion for the extraction of significant factors. In the case of determining the low homogeneity and complex latent structure of the scale, *item selection* was applied to simplify the latent structure and increase the homogeneity of the items of that scale. Factor saturations for each individual item were defined for both the initial and final versions of the scales by way of determining the common variance of the component, and with the percentage of the total explained variance. To determine the *reliability* of the scales, the *Cronbach's Alpha* internal reliability coefficients were calculated for each scale. To assess the *sensitivity* of the scales, the Kolmogorov–Smirnov (K–S) *goodness-of-fit* test, and *skewness* and *kurtosis* coefficients, were used. If the coefficient of the K–S test for some measure displayed significant deviations from the normal distribution, and if the sensitivity coefficients' *skewness* and *kurtosis* were in the range of $\pm1.00$, the measure was considered satisfactorily sensitive, which satisfied the criteria to use parametric statistical procedures (George and Malley 2010).

The *validity* of all ASI scales used in this study was determined in several different ways: through the identification of *gender differences* in the approval of sexism (*Student's t-test*); by using *correlation analysis* to assess the association of different measures of sexism (*Pearson's correlation coefficient*); through *multiple regression analysis* to identify the determinants of various sub-dimensions of sexism; by determining different *types of participants* according to their way of assessing sexism, applying the *K-means clustering procedure*; and through identifying differences in socio–demographic, religious, and occupational characteristics between groups of participants that belonged to these different types. A *K-means clustering* procedure, wherein participants were classified into a predetermined number of groups, was performed, in order to group participants together who had a very similar manner of approving sexism. At the same time, it was expected that groups of participants (*types*) would be significantly different in their approval of sexism. In these analyses, three different models with five, four and three clusters of participants were tested. The model with three clusters that was selected for this paper provided a simple and easy-to-interpret solution, and thus, was more acceptable. The *Chi-square test of independence* was used to determine the association between socio–demographic, religious and professional characteristics among participants. The data of the two category variables were arranged in a contingency table along the columns and rows. With each Chi-square test result, *Cramer's V* coefficient was calculated as a test *of the effect size* between variables, which is used to test any significant Chi-square result. Cramer's V coefficient is a form of correlation and is interpreted accordingly.

## 3. Results

In the PCA on the original scale items of HS, 10 items were found to be projected onto one of a total of two factors. In item selection, one item was rejected. The new, adapted HS scale contained 10 items and had high reliability ($\alpha = 0.92$), with the component explaining a 58.5% of the total variance (Table 1).

The PCA analysis determined that there were three dimensions on the BS scale items. In a two–step selection of the items, two items that disrupted the latent structure were discarded. The two latent dimensions were determined on the remaining items, which explained 49.9% of the total variance, and the BS scale had satisfactory reliability ($\alpha = 0.73$) (Table 2).

These results confirm that the BS construct was not unidimensional, and it was not well operationalized, so the process of adapting the BS scale was continued. In the further selection of items, two more items were rejected. Two separate scales of satisfactory homogeneity and reliability were designed, which were titled according to the content of the remaining items (Table 3).

**Table 1.** Structural and metric characteristics of the *Hostile Sexism* (*HS*) scale before and after item selection.

| No. | Item | F 1 | F 2 | F 1 * |
|-----|------|-----|-----|-------|
| Q_2 | Many women are actually seeking special favors, such as hiring policies that favor them over men, under the guise of asking for "equality." | 0.68 | 0.24 | −0.72 |
| Q_4 | Most women perform innocent remarks or acts as being sexist. | 0.74 | 0.15 | −0.75 |
| Q_5 | Women are too easily offended. | 0.78 | 0.09 | −0.79 |
| Q_7 | Feminists are seeking for women to have more power than men. ¥ | 0.60 | 0.47 | −0.68 |
| Q_10 | Most women fail to fully appreciate all that men do for them. | 0.69 | 0.11 | −0.70 |
| Q_11 | Women seek to gain power by gaining control over men. | 0.84 | 0.06 | −0.84 |
| Q_14 | Women exaggerate problems they have at work. | 0.81 | 0.13 | −0.81 |
| Q_15 | Once a woman gets a man to commit to her, she usually tries to put him on a tight leash. | 0.84 | 0.01 | −0.82 |
| Q_16 | When women lose to men in a fair competition, they typically complain about being discriminated against. | 0.83 | 0.12 | −0.84 |
| Q_18 | There are many women who get a kick out of teasing men by seeming sexually available and then refusing male advances. ¥ | 0.68 | 0.07 | −0.68 |
| Q_21 | Feminists are making entirely reasonable demands of men. (–) | −0.02 | −0.96 | – |
| | Eigen value | 5.67 | 1.28 | 5.85 |
| | % | 51.5 | 11.6 | 58.5 |
| | Alpha | | 0.89 | 0.92 |

Notes: ¥—in the Croatian version of the questionnaire, the positive form of the item was used instead of the original negative form; F n—factor saturation of item; F n *—factor saturation of item after item selection; Eigen value—inherent component variance; %—total percentage of explained variance; Alpha—Cronbach's Alpha coefficient; (–)—inverted item.

**Table 2.** Structural and metric characteristics of the *Benevolent Sexism* (*BS*) scale before and after item selection.

| No. | Item | F 1 | F 2 | F 3 | F 1 * | F 2 * |
|-----|------|-----|-----|-----|-------|-------|
| Q_1 | No matter how accomplished he is, a man is not truly complete as a person unless he has the love of a woman. | 0.20 | −0.57 | 0.43 | 0.25 | 0.73 |
| Q_3 | In a disaster, women ought not necessarily to be rescued before men. (–) | −0.29 | 0.48 | 0.35 | – | – |
| Q_6 | People are often truly happy in life without being romantically involved with a member of the other sex. (–) | 0.08 | 0.75 | −0.14 | 0.12 | −0.73 |
| Q_8 | Many women have a quality of purity that few men possess. | 0.69 | −0.09 | 0.16 | 0.70 | 0.14 |
| Q_9 | Women should be cherished and protected by men. | 0.08 | −0.05 | 0.64 | 0.24 | 0.36 |
| Q_12 | Every man ought to have a woman whom he adores. | 0.25 | −0.32 | 0.67 | 0.38 | 0.62 |
| Q_13 | Men are complete without women. (–) | −0.01 | 0.76 | −0.15 | 0.04 | −0.73 |
| Q_17 | A good woman should be set on a pedestal by her man. | 0.18 | −0.08 | 0.71 | – | – |
| Q_19 | Women, compared to men, tend to have a superior moral sensibility. | 0.78 | 0.04 | 0.03 | 0.77 | −0.03 |
| Q_20 | Men should be willing to sacrifice their own wellbeing in order to provide financially for the women in their lives. | 0.54 | −0.20 | 0.29 | 0.57 | 0.29 |
| Q_22 | Women, as compared to men, tend to have a more refined sense of culture and good taste. | 0.76 | 0.02 | 0.12 | 0.77 | 0.02 |
| | Eigen value | 2.19 | 1.86 | 1.84 | 2.27 | 2.22 |
| | % | 19.9 | 16.9 | 16.7 | 25.2 | 24.7 |
| | Alpha | | 0.74 | | | 0.73 |

Notes: F n—factor saturation of item; F n *—factor saturation of item after item selection; Eigen value—inherent component variance; %—total percentage of explained variance; (–)—inverted item; Alpha—Cronbach's Alpha coefficient.

The first adapted BS scale contained three items, its reliability was conditionally satisfactory ($\alpha = 0.69$), and it explained 61.9% of the common variance. The authors entitled this dimension of BS as *complementary gender differentiation* (BS_CGD). It integrated a concept of beliefs about the differences between women and men, with women being described as naturally gentler and purer than men (Glick and Fiske 1996; Connor et al. 2017). The

second adapted BS scale consisted of four items, had satisfactory reliability ($\alpha = 0.70$), and explained 52.7% of the common variance. The scale integrated a concept of romantic beliefs about role of a woman in an intimate relationship with man. The authors entitled this dimension of BS *heterosexual intimacy* (BS_HI), referring to romantic beliefs about a woman's ability to "complete" a man in their interdependent intimate relationship.

**Table 3.** Structural and metric characteristics of the two *Benevolent Sexism* (*BS*) scales.

| | BS Complementary Gender Differentiation (BS_CGD) Scale | |
|---|---|---|
| **No.** | **Item** | **F 1** |
| Q_8 | Many women have a quality of purity that few men possess. | −0.75 |
| Q_19 | Women, compared to men, tend to have a superior moral sensibility. | −0.82 |
| Q_22 | Women, as compared to men, tend to have a more refined sense of culture and good taste. | −0.79 |
| | Eigen value | 1.86 |
| | % | 61.9 |
| | Alpha | 0.69 |
| | **BS Heterosexual Intimacy (BS_HI) Scale** | |
| **No.** | **Item** | **F 1** |
| Q_1 | No matter how accomplished he is, a man is not truly complete as a person unless he has the love of a woman. | −0.80 |
| Q_6 | People are often truly happy in life without being romantically involved with a member of the other sex. (–) | 0.70 |
| Q_12 | Every man ought to have a woman whom he adores. | −0.69 |
| Q_13 | Men are complete without women. (–) | 0.71 |
| | Eigen value | 2.11 |
| | % | 52.7 |
| | Alpha | 0.70 |

Notes: F n—factor saturation of item; Eigen value—inherent component variance; %—total percentage of explained variance; Alpha—Cronbach's Alpha coefficient; (–)—inverted item.

Table 4 shows the descriptive characteristics of all the validated scales.

**Table 4.** Gender differences and sensitivity of validated ASI scales.

| Variable | Women (N = 447) | | | | | Men (N = 438) | | | | | *t*-Test | *p* = |
|---|---|---|---|---|---|---|---|---|---|---|---|---|
| | Mean ± SD | MED | SKEW | KURT | KS D | Mean ± SD | MED | SKEW | KURT | KS D | | |
| **ASI_TOTAL** | 2.24 ± 0.76 | 2.26 | −0.03 | −0.20 | 0.03 | 2.76 ± 0.70 | 2.84 | −0.51 | 0.10 | 0.06 | 10.21 | <0.001 |
| **HS** | 1.82 ± 1.02 | 1.70 | 0.33 | −0.55 | 0.08 * | 2.72 ± 1.13 | 2.70 | −0.17 | −0.44 | 0.04 | 12.32 | <0.001 |
| **BS_TOTAL** | 2.73 ± 0.90 | 2.78 | −0.26 | −0.42 | 0.05 | 2.83 ± 0.72 | 2.89 | −0.35 | 0.06 | 0.06 | 1.85 | 0.07 |
| **BS_CGD** | 2.32 ± 1.24 | 2.33 | −0.03 | −0.75 | 0.08 * | 1.95 ± 1.07 | 2.00 | 0.05 | −0.59 | 0.07 * | 4.76 | <0.001 |
| **BS_HI** | 2.86 ± 1.17 | 3.00 | −0.30 | −0.48 | 0.09 * | 3.23 ± 1.00 | 3.25 | −0.47 | 0.02 | 0.08 * | 5.06 | <0.001 |

Notes: Mean ± SD—arithmetic mean and standard deviation; MED—median; SKEW—coefficient of the asymmetry of distribution results; KURT—coefficient of the kurtosis of distribution results; KS D—Kolmogorov–Smirnov goodness-of-fit test; *—significant K–S D test coefficient; *t*-test—*t*-test coefficient; *p* =—significance of the *t*-test coefficient.

The metric characteristic of the *sensitivity* of the scales was satisfactory (*skewness* and *kurtosis coefficients* of all scales were in the range of ±1.00), which enabled the use of parametric statistical procedures. Gender differences in the endorsement of sexism were found in four of the five presented measures. Only for the total measure of BS was no gender difference identified. Men scored higher than women on the overall ASI measure, the HS measure, and the BS_HI measure. Women scored higher than men on the BS_CGD measure. Due to the identified gender differences, all further statistical analyses were performed separately for each gender group.

Consequently, correlations were calculated for all ASI measures for both women and men separately (Table 5).

**Table 5.** Correlations between the validated ASI measures.

| Variable | Women | | | | |
|---|---|---|---|---|---|
| | ASI | HS | BS | BS_CGD | BS_HI |
| **ASI_TOTAL** | 1.00 | 0.84 ** | 0.73 ** | 0.59 ** | 0.56 ** |
| **HS** | 0.84 ** | 1.00 | 0.24 ** | 0.21 ** | 0.14 * |
| **BS_TOTAL** | 0.73 ** | 0.24 ** | 1.00 | 0.80 ** | 0.83 ** |
| **BS_CGD** | 0.59 ** | 0.21 ** | 0.80 ** | 1.00 | 0.38 ** |
| **BS_HI** | 0.56 ** | 0.14 * | 0.83 ** | 0.38 ** | 1.00 |
| Variable | Men | | | | |
| | ASI | HS | BS | BS_CGD | BS_HI |
| **ASI_TOTAL** | 1.00 | 0.87 ** | 0.52 ** | 0.39 ** | 0.37 ** |
| **HS** | 0.87 ** | 1.00 | 0.04 | 0.07 | −0.04 |
| **BS_TOTAL** | 0.52 ** | 0.04 | 1.00 | 0.67 ** | 0.80 ** |
| **BS_CGD** | 0.39 ** | 0.07 | 0.67 ** | 1.00 | 0.16 ** |
| **BS_HI** | 0.37 ** | −0.04 | 0.80 ** | 0.16 ** | 1.00 |

Notes: *—$p < 0.01$ coefficient; **—$p < 0.001$ coefficient.

For both samples, significant moderate-to-high correlations of the overall ASI measure with all HS and BS measures were found. However, significant but low positive correlations of HS with all BS measures were found among women. A low correlation between BS_CGD and BS_HI was also identified, which explained only 14.4% of the common variance. By contrast, no statistically significant correlations between the HS and BS measures were found among men, and the positive correlation between the two BS sub–dimensions proved very low ($r = 0.16$). Because of the low correlation between two BS measures (in both subsamples), two separate BS measures, instead of a single BS measure, were used.

Subsequent analyses were only conducted with the best estimated determinants of HS and BS measures. Multiple regression analyses were performed to identify determinants of HS, BS_CGD and BS_HI (Tables 6 and 7).

**Table 6.** Determinants of the *sexism* measures among women.

| Variable | Hostile Sexism | | | BS Complementary Gender Differentiation | | | BS Heterosexual Intimacy | | | Tolerance |
|---|---|---|---|---|---|---|---|---|---|---|
| | BETA | *t* (425) | *p* = | BETA | *t* (426) | *p* = | BETA | *t* (428) | *p* = | |
| *Intercept* | — | 8.31 | <0.001 | — | 4.11 | <0.001 | — | 7.44 | <0.001 | — |
| Age | −0.37 | −3.22 | 0.001 | −0.07 | −0.67 | 0.50 | −0.23 | −2.10 | 0.036 | 0.17 |
| Education degree | −0.11 | −1.49 | 0.14 | −0.04 | −0.52 | 0.61 | −0.06 | −0.79 | 0.43 | 0.38 |
| Professional status | 0.01 | 0.15 | 0.88 | −0.22 | −2.48 | 0.01 | −0.06 | −0.70 | 0.48 | 0.25 |
| Place of growing up | 0.04 | 0.83 | 0.41 | −0.02 | −0.48 | 0.63 | −0.08 | −1.64 | 0.10 | 0.88 |
| Command duty | −0.04 | −0.81 | 0.42 | −0.03 | −0.70 | 0.49 | −0.04 | −0.82 | 0.42 | 0.84 |
| Years of military service | −0.21 | −1.90 | 0.06 | 0.08 | 0.73 | 0.46 | −0.01 | −0.14 | 0.89 | 0.18 |
| Faith importance | 0.04 | 0.67 | 0.51 | 0.13 | 2.25 | 0.02 | 0.19 | 3.33 | 0.001 | 0.60 |
| Religious practice | −0.04 | −0.60 | 0.55 | 0.08 | 1.49 | 0.14 | 0.06 | 1.00 | 0.32 | 0.61 |
| R | | 0.24 | | | 0.40 | | | 0.38 | | |
| R² | | 0.06 | | | 0.16 | | | 0.15 | | |
| Adjusted R² | | 0.04 | | | 0.14 | | | 0.13 | | |
| F (df) | | 3.20 (8,423) | | | 9.95 (8,426) | | | 9.18 (8,426) | | |
| *p* = | | 0.002 | | | <0.001 | | | <0.001 | | |

Notes: BETA—regression coefficient; R—multiple correlation coefficient; R²—coefficient of determination; Adjusted R²—adjusted multiple determination coefficient; F—coefficient of significance of multiple regression; df—degrees of freedom; $p$ =—level of coefficient of significance of multiple regression.

**Table 7.** Determinants of the three validated *sexism* measures among men.

| Variable | Hostile Sexism | | | BS Complementary Gender Differentiation | | | BS Heterosexual Intimacy | | | Tolerance |
|---|---|---|---|---|---|---|---|---|---|---|
| | BETA | *t* (425) | *p* = | BETA | *t* (426) | *p* = | BETA | *t* (428) | *p* = | |
| *Intercept* | – | 6.84 | <0.001 | – | 2.82 | <0.001 | – | 5.15 | <0.001 | – |
| Age | −0.10 | −0.81 | 0.42 | −0.04 | −0.31 | 0.76 | −0.13 | −1.05 | 0.30 | 0.13 |
| Educational level | −0.09 | −1.25 | 0.21 | 0.06 | 0.78 | 0.43 | 0.05 | 0.63 | 0.53 | 0.42 |
| Professional status | −0.17 | −1.93 | 0.054 | −0.14 | −1.47 | 0.14 | −0.02 | −0.23 | 0.82 | 0.27 |
| Place of growing up | −0.09 | −1.86 | 0.063 | −0.01 | −0.17 | 0.86 | −0.00 | −0.00 | 1.00 | 0.93 |
| Command duty | −0.06 | −1.27 | 0.21 | 0.04 | 0.69 | 0.49 | 0.07 | 1.42 | 0.16 | 0.86 |
| Years of military service | 0.16 | 1.36 | 0.17 | 0.00 | 0.02 | 0.99 | −0.21 | −1.74 | 0.08 | 0.14 |
| Faith importance | −0.03 | −0.59 | 0.56 | 0.04 | 0.70 | 0.48 | 0.27 | 4.52 | <0.001 | 0.63 |
| Religious practice | 0.00 | 0.02 | 0.98 | 0.12 | 1.89 | 0.06 | 0.07 | 1.15 | 0.25 | 0.62 |
| R | 0.43 | | | 0.21 | | | 0.33 | | | |
| $R^2$ | 0.18 | | | 0.04 | | | 0.11 | | | |
| Adjusted $R^2$ | 0.17 | | | 0.02 | | | 0.09 | | | |
| F (df) | 11.37 (8,404) | | | 2.27 (8,404) | | | 6.25 (8,404) | | | |
| *p* = | <0.001 | | | 0.02 | | | <0.001 | | | |

Notes: BETA—regression coefficient; R—multiple correlation coefficient; $R^2$—coefficient of determination; Adjusted $R^2$—adjusted multiple determination coefficient; F—coefficient of significance of multiple regression; df—degrees of freedom; *p* =—level of coefficient of significance of multiple regression.

The regression models for women were significant for all three sexism measures. However, all the predictors accounted for 6% of the variance of HS, 16% of the variance of BS_CGD, and 15% of the variance of BS_HI. The multicollinearity among predictors were not detected (all *tolerance coefficients* were above 0.10). The *age* variable was the negative predictor of HS. The *professional status* variable was negative, whereas *faith importance* was a positive predictor of BS_CGD. In the prediction of variance of BS_HI, *age* was negative, whereas *faith importance* was a positive predictor. All accounted coefficients were low.

Among men, all the predictors accounted for 18% of the variance of HS, 4% of the variance of BS_CGD, and 11% of the variance of BS_HI only. In the prediction of both HS and BS_CDG, no significant independent contributions of any of the predictors were identified. The *faith importance* variable was the only predictor of BS_HI, and was positive.

Previously determined gender differences in different measures of sexism allowed for *classification* of participants. Each participant from each gender group was assigned to one of the three possible clusters via the *K-means clustering procedure*. The frequencies and descriptive characteristics for each cluster and for both women and men are shown in Table 8.

Among women, three *types* of sexism endorsement were identified. The first group constituted 25.3% of the female subsample and included women with low values for HS, BS_CGD, and BS_HI. This type of woman was named *egalitarian*. The second group constituted 28.1% of the female subsample and included women whose values for all three sexism measures were higher than those of the first group. More precisely, the second group included women with low values for HS and BS_CGD, but with high values for BS_HI. This type of woman was named *moderate egalitarian*. The third group constituted 46.5% of the female subsample and included women with moderately low values for HS and moderately high values for BS_CGD and BS_HI. Given their assessments, they generally endorsed traditional beliefs about women. Therefore, this type of woman was named *traditional*.

Three *types* of sexism endorsement were identified among men, too. The first group constituted 30.7% of the male subsample and included men with low values for HS and BS_CGD, and with moderate values for BS_HI. This type of man was named *moderate egalitarian*. The second group constituted 44.6% of the total male subsample and included men with moderate values for HS and BS_CGD, and with high values for BS_HI. This type of man was named *traditional*. The third group constituted 24.6% of the male subsample and included men with very high values for HS, very low values for BS_CGD, and moderate–

high values for BS_HI. They actually endorsed more traditional but also hostile attitudes towards women. This type of man was named *hostile*.

**Table 8.** Types of sexism assessment among women and men.

| Variable | Women | | | F | *p* = |
| --- | --- | --- | --- | --- | --- |
| | *Egalitarian* (N = 110) | *Moderate Egalitarian* (N = 122) | *Traditional* (N = 202) | | |
| | **Mean ± SD** | **Mean ± SD** | **Mean ± SD** | | |
| **HS** | 1.40 ± 0.92 | 1.71 ± 0.96 | 2.09 ± 1.03 | 18.51 | <0.001 |
| **BS_CGD** | 1.24 ± 0.85 | 1.51 ± 0.69 | 3.40 ± 0.67 | 423.33 | <0.001 |
| **BS_HI** | 1.39 ± 0.67 | 3.50 ± 0.72 | 3.27 ± 0.90 | 252.57 | <0.001 |
| Variable | Men | | | F | *p* = |
| | *Moderate Egalitarian* (N = 126) | *Traditional* (N = 183) | *Hostile* (N = 101) | | |
| | **Mean ± SD** | **Mean ± SD** | **Mean ± SD** | | |
| **HS** | 1.56 ± 0.76 | 2.84 ± 0.71 | 3.94 ± 0.64 | 318.81 | <0.001 |
| **BS_CGD** | 1.33 ± 0.80 | 2.74 ± 0.73 | 1.27 ± 0.79 | 177.64 | <0.001 |
| **BS_HI** | 2.85 ± 1.03 | 3.69 ± 0.75 | 2.72 ± 0.90 | 53.06 | <0.001 |

Notes: Mean ± SD—arithmetic mean and standard deviation; F—One-way ANOVA coefficient; *p* =—significance of the One-way ANOVA coefficient.

Differences among types of sexism endorsement for women and men are depicted in Figure 1.

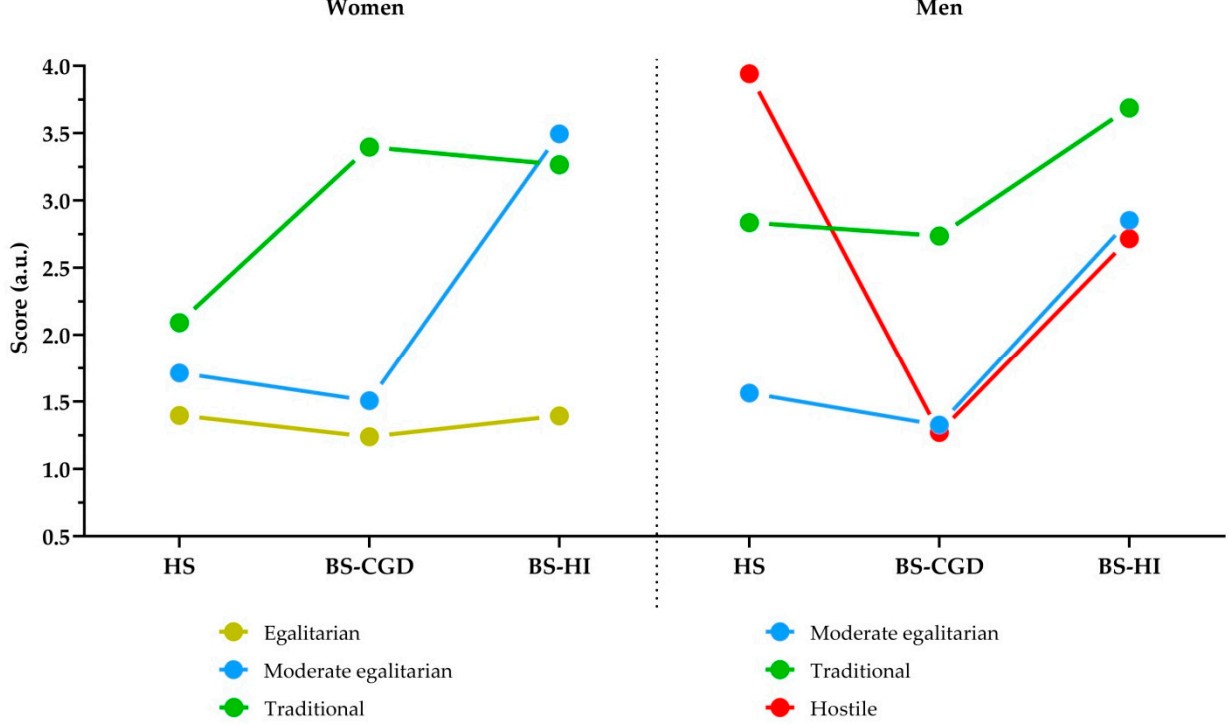

**Figure 1.** Types of sexism endorsement among women and men.

Figure 1 shows differences in sexism endorsement types within both subsamples, but also a gender differences in the structure of identified types. To determine the individual characteristics of members of the groups of different types of sexism endorsement, a non–parametric *Chi-square* test was used, both for women (Table 9) and men (Table 10).

**Table 9.** Differences in women's *sexism endorsement types* according to personal characteristics.

| Variable | Category | Egalitarian (N = 110) | Moderate Egalitarian (N = 122) | Traditional (N = 202) | Chi-Square—Test of Independence | |
|---|---|---|---|---|---|---|
| | | % | % | % | | |
| Age | 20–29 years | 7.4 | 21.7 | 42.4 | Chi-square | 47.56 |
| | 30–39 years | 30.6 | 24.2 | 21.7 | (df) | (426) |
| | 40–49 years | 34.3 | 31.7 | 22.2 | Cramer's V | 0.24 |
| | 50–59 years | 27.8 | 22.5 | 13.6 | *p* = | <0.001 |
| Educational level | High school | 40.0 | 49.2 | 68.8 | Chi-square | 27.19 |
| | | | | | (df) | (434) |
| | Bachelor's degree or higher | 60.0 | 50.8 | 31.2 | Cramer's V | 0.25 |
| | | | | | *p* = | <0.001 |
| Professional status | Soldier (OR1–OR3) | 14.7 | 23.8 | 47.3 | Chi-square | 51.83 |
| | NCO (OR4–OR9) | 30.3 | 36.1 | 30.3 | (df) | (432) |
| | Junior CO (OF1–OF2) | 33.0 | 27.0 | 15.9 | Cramer's V | 0.25 |
| | Senior CO (OF3–OF5) | 22.0 | 13.1 | 6.5 | *p* = | <0.001 |
| Place of growing up | Village | 26.9 | 40.2 | 48.0 | Chi-square | 17.83 |
| | Very small town | 17.6 | 20.5 | 15.8 | | |
| | Small town | 20.4 | 16.4 | 15.8 | (df) | (432) |
| | Town | 16.7 | 9.8 | 11.9 | Cramer's V | 0.14 |
| | Big town | 18.5 | 13.1 | 8.4 | *p* = | 0.023 |
| Years of military service | 23 years or more | 44.3 | 40.2 | 30.2 | | |
| | 18–22 years | 10.4 | 9.0 | 5.0 | Chi-square | 43.90 |
| | 13–17 years | 9.4 | 9.8 | 4.0 | | |
| | 8–12 years | 20.8 | 18.0 | 14.1 | (df) | (427) |
| | 3–7 years | 8.5 | 9.0 | 13.6 | Cramer's V | 0.23 |
| | 0–2 years | 6.6 | 13.9 | 33.2 | *p* = | <0.001 |
| Faith importance | (*completely irrelevant*) 1–2 | 8.3 | 3.2 | 3.5 | Chi-square | 26.67 |
| | 3–4 | 5.5 | 3.3 | 4.0 | | |
| | 5–6 | 16.7 | 12.3 | 11.0 | (df) | (430) |
| | 7–8 | 29.6 | 24.6 | 25.0 | Cramer's V | 0.18 |
| | 9–10 (*very important*) | 39.8 | 56.5 | 56.5 | *p* = | 0.09 |
| Religious practice | 1 (*almost never*)– 2 (*rare*) | 23.8 | 18.8 | 14.9 | Chi-square | 23.15 |
| | 3 (*per year*)– 4 (*for religious festivals*) | 33.0 | 21.3 | 25.4 | (df) | (432) |
| | 5 (*once a month*) | 20.2 | 18.0 | 22.4 | Cramer's V | 0.16 |
| | 6 (*per week*)– 7 (*more than weekly*) | 23.0 | 41.8 | 37.3 | *p* = | 0.026 |

Notes: Chi-square—Chi-square test of the independence coefficient; df—degrees of freedom; Cramer's V—Cramer's V coefficient; *p* =—significance of the Chi-square coefficient.

**Table 10.** Differences in men's *sexism endorsement types* according to personal characteristics.

| Variable | Category | Moderate Egalitarian (N = 126) | Traditional (N = 183) | Hostile (N = 101) | Chi-Square—Test of Independence | |
|---|---|---|---|---|---|---|
| | | % | % | % | | |
| Age | 20–29 years | 12.9 | 32.6 | 42.4 | Chi-square | 32.75 |
| | 30–39 years | 24.2 | 26.9 | 24.2 | (df) | (398) |
| | 40–49 years | 40.3 | 27.4 | 26.3 | Cramer's V | 0.20 |
| | 50–59 years | 22.6 | 13.1 | 7.1 | *p* = | <0.001 |
| Educational level | High school | 61.9 | 76.0 | 86.1 | Chi-square | 17.74 |
| | | | | | (df) | (410) |
| | Bachelor's degree or higher | 38.1 | 24.0 | 13.9 | Cramer's V | 0.21 |
| | | | | | *p* = | <0.001 |
| Professional status | Soldier (OR1–OR3) | 20.6 | 43.2 | 50.5 | Chi-square | 36.30 |
| | NCO (OR4–OR9) | 43.7 | 36.6 | 40.6 | (df) | (410) |
| | Junior CO (OF1–OF2) | 18.3 | 10.9 | 6.9 | Cramer's V | 0.21 |
| | Senior CO (OF3–OF5) | 17.5 | 9.3 | 2.0 | *p* = | <0.001 |
| Place of growing up | Village | 30.2 | 45.4 | 46.5 | Chi-square | 13.03 |
| | A very small town | 19.8 | 17.5 | 19.8 | | |
| | Smaller town | 28.6 | 25.1 | 18.8 | (df) | (410) |
| | City | 11.1 | 7.7 | 6.9 | Cramer's V | 0.18 |
| | Big city | 10.3 | 4.4 | 7.9 | *p* = | 0.13 |
| Years of military service | 23 years or more | 50.0 | 31.8 | 20.6 | | |
| | 18–22 years | 8.1 | 5.1 | 12.4 | Chi-square | 37.25 |
| | 13–17 years | 8.9 | 6.8 | 5.2 | | |
| | 8–12 years | 17.7 | 23.3 | 18.6 | (df) | (397) |
| | 3–7 years | 8.9 | 14.2 | 17.5 | Cramer's V | 0.22 |
| | 0–2 years | 6.5 | 18.8 | 25.8 | *p* = | <0.001 |
| Faith importance | (completely irrelevant) 1–2 | 14.3 | 4.4 | 11.9 | Chi-square | 23.90 |
| | 3–4 | 7.2 | 4.9 | 3.0 | | |
| | 5–6 | 17.4 | 12.6 | 12.9 | (df) | (409) |
| | 7–8 | 20.6 | 29.7 | 23.8 | Cramer's V | 0.17 |
| | 9–10 (very important) | 40.5 | 48.4 | 48.5 | *p* = | 0.16 |
| Religious practice | 1 (almost never)– 2 (rare) | 32.6 | 19.2 | 26.7 | Chi-square | 17.28 |
| | 3 (per year)– 4 (for holidays) | 29.3 | 36.6 | 36.6 | (df) | (410) |
| | 5 (once a month) | 11.9 | 14.8 | 13.9 | Cramer's V | 0.15 |
| | 6 (per week)– 7 (more than weekly) | 26.2 | 29.5 | 22.7 | *p* = | 0.14 |

Notes: Chi-square—Chi-square test of the independence coefficient; df—degrees of freedom; Cramer's V—Cramer's V coefficient; *p* =—significance of the Chi-square coefficient.

Among women, the results revealed significant correlations between the selected sociodemographic, professional, and religious characteristics of members of different sexism *types* for all the variables used, except for *faith importance*. A *large effect* was revealed for the variables *age*, *educational level*, *professional status* and *years of military service*, while a *medium effect* of the relationship was confirmed for *religious practice* and *place of growing up*. Over 50% of *egalitarian* and *moderate egalitarian* women were aged 40 and over, whereas the majority of *traditional* women (42.4%) were aged 20 to 29. The majority of *traditional* women completed *high school* (68.8%), while among *egalitarian* and *moderate egalitarian* women, the majority had *completed a bachelor's degree or higher* (60% vs. 50.8%). Among *traditional* women, the majority were *soldiers* (47.3%) and the fewest were *senior officers* (6.5%). Most of the *egalitarian* women were *officers* (33% *junior officers* and 22% *senior officers*). Although most of the *moderate egalitarian* women were NCOs (36.1%), it was evident that those women

were equally represented in all professional groups. Most of *traditional* women grew up in *rural areas* (48%), similarly to *moderate egalitarian* women (40.2%), while the highest proportion of *egalitarian* women grew up in *bigger towns*. There was also a significant difference in terms of years of military service. Most of the *traditional* women served for *up to two years* (33.2%), whereas most of the *moderately egalitarian* and *egalitarian* women served for more than 23 years (40.2% vs. 44.3%). The majority of *traditional* women attended religious ceremonies *once a week or more often* (37.3%), and the minority of *traditional* women (as well as among women of other types) *almost never attended them* (14.9%). In contrast, the highest proportions of women who attended religious ceremonies *once a year* or *only for religious holidays* (33%), and of those who *did not attend them at all* (23.8%), were found among *egalitarian* women.

Among men, the results indicated significant association of selected socio–demographic and professional characteristics of members of different sexism endorsement *types*; however this was not the case for the *religious* variables used. A *large effect* was found for the *years of military service*, and a *medium effect* of the relationship was found for the *age, educational level* and *professional status*. More than half of the *moderate egalitarian* men were *aged 40 and over* (60%), whereas the majority of *traditional* and *hostile* men were *aged 20 to 29* (32.6% vs. 42.4%). Although the proportion of those who had at least completed a *bachelor's degree* was significantly lower in all three sexism types of men, more highly educated men were much less represented among *hostile* men (13.9%) than among *moderately egalitarian* and *traditional* men (38.1% vs. 24%). A large number of *moderate egalitarian* men were NCOs (43.7%) and officers (35%), whereas the majority of *traditional* men were soldiers (43.2%) and NCOs (36.6%). Among *hostile* men, the vast majority were soldiers and NCOs (50.5%; 40.6%), and only 8.9% were officers. The majority of *moderate egalitarian* and *traditional* men served *for more than 23 years* (50% vs. 36%). In contrast, most of the *hostile* men served for *up to two years* (25.6%). The associations of the *place of growing up*, *faith importance* and *religiosity* with the *sexist types* were not identified.

## 4. Discussion

There are several major findings from this study: (1) the ASI scales had satisfactory metric properties, and the basic dimensions of ambivalent sexism were confirmed; (2) a moderately high endorsement of sexism and significant gender differences were found. Men endorsed HS and BS_HI to a greater extent, and women BS_CGD; (3) the predictor set explained a very small amount of variance of the criterion variables. Significant independent contributions were only found for *age* and *faith importance*; (4) three *sexism types* were singled out in both subsamples: *egalitarian, moderate egalitarian* and *traditional* women, and *moderate egalitarian, traditional* and *hostile* men; (5) *traditional* women, compared to *egalitarian* and *moderate egalitarian* women, were more likely to be younger, of a lower educational level and professional status, come from rural areas, have served a shorter time in the military, and be more religious; (6) *hostile* men were more likely than *moderate egalitarian* and *traditional* men to be younger, of *lower educational level and professional status*, and have served a shorter time in the *military*. These findings require a more precise and detailed interpretation and will be further presented.

### 4.1. Validation of Sexism Scales

The validation of ASI determined the unidimensional structure of the HS scale and the bidimensional structure of the BS scale. It was confirmed that three sexist scales (HS, BD_CGD and BS_HI) had satisfactory measurement characteristics. With regard to theoretical assumptions, the present study confirmed only two dimensions of the BS construct, namely *complementary gender differentiation*, and *heterosexual intimacy*; however, the third *protective paternalism* dimension was lacking. These differences in the structure of the BS could be attributed to cultural differences in the understanding of gender roles, which could lead to differences in the meaning and importance of individual items of the original scale in various cultures (McHugh and Frieze 1997). Furthermore, Mikołajczak

and Pietrzak (2015) suggested the possibility that within the same cultural context, there are specific subgroups that have their own system of gender-related attitudes in which sexism can be manifested differently; these could apply to the military setting that was investigated in this study. More specifically, the dimension of *protective paternalism* refers to traditional beliefs about the need for protective behavior of men towards women, which is incompatible with the characteristics of the military profession and the modern military. Therefore, it could be assumed that with the professionalization of the armed forces and with the greater participation of women in active service, there has been a change in attitudes about the need to protect women in the military. We assume that this was due to the fact that women voluntarily chose the military profession. This implies a high level of military training, and includes a high level of adoption of protection skills. The study findings confirmed the cultural specifics of ASI and suggested the justification for employing theoretically assumed dimensions of BS in other studies.

Overall, the participants in the present study endorsed sexism from a *low* to *moderate* degree. The results of endorsing both HS and BS in women and men confirmed the theoretical assumption about the ambivalent nature of sexism. Glick and Fiske (2011) associated consistent research findings on women endorsing sexism with the process of social learning of sexist beliefs in society. The endorsement of sexism among women was explained by means of the *system-justification theory* (Glick and Fiske 1996, 2001). It was assumed that women, as well as members of other underprivileged social groups, tend to justify existing social inequalities by accepting the belief in the deserved place of their own social group in the social hierarchy, or perceive that the existing *status quo* is actually fair (Connelly and Heesacker 2012). These interpretations could, likewise, be applied to the results of women in the present study.

*4.2. Gender Differences in Sexism Endorsement*

This study finding revealed gender differences. Men had more pronounced hostile sexism and sexist beliefs that glorify the role of women in a close heterosexual relationship (BS_HI), while women had more pronounced sexist beliefs that idealize their stereotypical gender traits (BS_CGD). These findings are in line with previous research (Glick and Fiske 1996; Masser and Abrams 1999; Glick et al. 2000; Fiske and North 2014; Mikołajczak and Pietrzak 2014; Mastari et al. 2019). It could be assumed that some men direct their hostility towards women because they have deviated from their traditional gender roles by opting for a military profession, and have therefore begun to "threaten" men's power and status in the military.

The identified gender differences in the endorsement of BS could also be attributed to the institutional (military) context. According to the theoretical assumptions, individuals who generally strongly approve sexist beliefs about women's weaker competencies compared to men are more likely to believe in women's dependence on men's protection. Additionally, they are more inclined to endorse traditional gender relations (Glick et al. 2000; Gaunt 2013), and therefore, the division of labor corresponding to them (Barreto and Ellemers 2013; Lorenzi-Cioldi and Kulich 2015). The BS_CGD dimension refers to the beliefs about differences in the traits and roles of women and men, where women are idealized and portrayed as naturally gentler, more sensitive, and morally purer than men (Connor et al. 2017; Hammond et al. 2017; Hammond and Overall 2017). Conversely, the military highly values qualities that are traditionally ascribed to men, but not to women (e.g., strength, courage, determination and rationality). Therefore, it could be expected that women who comparatively strongly approve BS_CGD will be reluctant to accept jobs and tasks in the military that they consider "not appropriate for women". In contrast, men who strongly approve of these beliefs are expected to be less inclined to reward women who have accepted "men–only" duties and jobs in the military, and to reward women who have chosen "women-suitable" jobs in the military. Thus, by approving seemingly subjective benevolent beliefs that idealize stereotypical traits of women—as well as beliefs about the non-compliance of these traits with the required standards of the military profession—

women in the military are directed towards occupational specialization. As women in the present study endorsed BS_CGD to a higher degree than men, it is reasonable to assume that military women will not only be likely to accept, but rather, more approving of the exclusion of women from certain duties and roles, especially combat ones. However, this, in fact, perpetuates gender inequality in the military; this is because women are prevented and/or restricted from performing precisely those duties that are a prerequisite for prestige, professional remuneration and faster rank advancement (Heinecken 2017). Further, it is important to emphasize that the study findings suggested a generally lower endorsement of BS_CGD compared to BS_HI in the total sample, which could be attributed to the specificities of the military profession. Namely, according to theoretical assumptions, beliefs about *complementary gender differences*—together with *protective paternalism*—lead women to be perceived as requiring the protection of men (Lorenzi-Cioldi and Kulich 2015), which conflicts with the expectations of their active-duty service.

The BS_HI sub-dimension refers to romantic beliefs about a woman's natural role in "complementing" a man in their interdependent heterosexual relationship (Connor et al. 2017; Hammond et al. 2017; Hammond and Overall 2017). The obtained results showed that in the total sample, these sexist beliefs were endorsed to a moderately high degree and that regardless of minor gender differences, men still endorsed them to a significantly higher degree than women. In addition, military men endorsed them much more than BS_CGD, and even more than HS. According to theoretical assumptions, individuals who strongly approve romantic beliefs about heterosexual intimacy are more likely to develop loyalty relationships that correspond to traditional, patriarchal patterns of female–male relationships (Glick and Fiske 1996, 2001). It is reasonable to assume that these beliefs contribute to gender inequality and the maintenance of the existing traditional gender hierarchy in the military.

The study findings generally suggested a high occurrence of sexism in the surveyed military population. This was not only confirmed by a significant gender gap in the average values of HS, but also in the average values of all sexism measures for both subsamples. In comparison to the results of a cross-cultural study (Glick et al. 2000), it is evident that the obtained values of military participants are comparable to the average values of the majority of traditional national samples (cultures), despite the large number of (previously) undertaken political activities, and the fact that the present study was conducted two decades afterwards. Moreover, an extra indicator of the "traditionalism" of the military work environment (in terms of sexism) is the result depicting low gender differences in the endorsement of BS. Glick and Fiske (2011) argued that due to higher exposure to sexism, women are not only prone to stronger rejection of HS, but also to greater acceptance of BS. Stronger endorsement of BS is considered to be a form of women's self-protection, driven by the need to gain the protection, admiration and sympathy of men and/or avoid their hostility (Glick et al. 2000; Fischer 2006). The obtained values of the HS and BS among women were congruent with these assertions, and also led to the conclusion of a higher presence of sexism in the Croatian military setting.

*4.3. Association of Sexism Measures*

In the present study, low positive correlations of HS with both BS measures were found among women, while no associations were found for men whatsoever. This finding was also in line with expectations, as previous research showed higher correlation between HS and BS among women and among those individuals who approve sexism to a lesser extent, i.e., for whom a low association between HS and BS are characteristic of sexist and traditional cultures (Glick et al. 2000; Zakrisson et al. 2012). Although low, the association between HS and BS among women confirmed the theoretical assumption of the complementarity of these two sexist ideologies, which effectively maintain the existing unequal gender relations in the military. At the same time, the lack of correlation between HS and BS among men further confirmed how strongly they endorse sexism towards women. While such correlation patterns might indicate a lack of awareness of the complementary nature

of sexism among men in the military, they might also point to their highly polarized view of women (Zakrisson et al. 2012). The finding of relatively high levels of endorsement of sexism among men—combined with the finding of a lack of intercorrelations—suggest that different types of male sexist attitudes may be directed at different types of women: HS towards women who deviated from their traditional gender roles, and BS towards women who adhered to them (Glick et al. 1997; Mikołajczak and Pietrzak 2015).

*4.4. Determinants of Ambivalent Sexism*

By identifying the determinants of ambivalent sexism measures, neither its expected impact nor its expected effect size were confirmed. In the results obtained for both subsamples, as for all sexism measures, minor amounts of common variance were explained. Furthermore, the chosen predictors explained sexism to a greater extent in the female subsample. The regression model better explained individual differences in the endorsement of BS_CGD and BS_HI among women, and individual differences in the endorsement of HS among men. For the military women, *age* was a negative predictor of HS, and was the only one. Furthermore, *professional status* and *faith importance* were predictors of BS_CGD, with stronger endorsement of BS_CGD among women of lower *professional status* and with higher *faith importance*. Similarly, *age* and *faith importance* were predictors of BS_HI among women. Thus, younger women, as well as women who felt that religion was more important to them, strongly endorsed BS_HI. Among men, *faith importance* had the strongest contribution in explaining the variance of HS. Therefore, HS was more strongly supported by men who attached more importance to religion.

In addition to *gender*, the most important predictors of attitudes towards gender roles were *age*, *educational level* and *place of growing up* (as suggested by Brajdić-Vuković et al. 2007). However, only a predictive effect of *age* was found in the present study. It could be assumed that the current findings are based on the common values of the overall military population. Values are adopted through socialization from the earliest age, and internalized values remain relatively stable throughout one's lifetime (Šverko et al. 2007). Therefore, it is justified to assume that the overall military sample generally included a large number of individuals who approved traditional values, emphasizing respect for authority, family values and acceptance of gender inequality. This is congruent with the results of previous research conducted in Croatia, which, despite showing a higher presence of egalitarian attitudes among Croatian citizens, warned of the emergence of re-traditionalization of gender roles (Sinovčić et al. 2016; Štimac Radin 2014). Moreover, the finding that the chosen predictors more strongly predicted BS among women, but HS among men, was attributed to the fact that sexism towards women was examined in this study. According to the *theory of social identity* (Tajfel and Turner 1986), a more positive evaluation and favoring of one's *own group*, and a negative evaluation and discrimination *of the external group*, was expected. As women in this study expressed attitudes towards *their own group*, it was reasonable to assume that they would be more likely to favor subjectively positive benevolent beliefs about women. Conversely, it was expected that men, in expressing attitudes towards members of the *external group*, would be more inclined to approve of hostile beliefs about the inferiority of women and generally insufficient competencies of women.

Furthermore, *age* only proved to be a negative predictor of HS and BS_HI for women. These unexpected results could point to gender differences in the socialization of gender attitudes. Stronger endorsement of sexism among younger women may mean that after joining active-duty service, they began to question traditionally prescribed gender norms and behaviors. At the same time, with the development of their careers and upon gaining military experience, some women are able to abandon the traditional view of women in the military, quit professional specialization in administrative or health duties, and realize career aspirations in combat tasks and duties. As a predictive effect of *age* was not found among men, it could be assumed that their strongly internalized social patterns of gender roles in the traditional, masculine work environment were more difficult to change, or that military culture could have further bolstered them (Heinecken 2017).

The findings about *faith importance* as a predictor of both dimensions of BS among women and BS_HI among men are in line with previous research (Burn and Busso 2005; Glick et al. 2002; Hannover et al. 2018; Mikołajczak and Pietrzak 2014). Religiosity is considered one of the fundamental values and reflects a commitment to religious beliefs (Brown 1996). Therefore, the values that religion and the Church promote also determine the ways in which believers generally perceive and interpret social relations. Croatia is a country where most people express their affiliation with Catholicism (Nikodem and Zrinščak 2019); thus, the findings from the present study could be linked to the norms and practices of the *Catholic Church* in Croatia. Its significance and public prominence increased significantly in the 1990s, amid the ongoing social system transformations and founding of the military. The Catholic Church strongly promotes traditional values, family life and the role of mothers in raising children; therefore, a greater acceptance of beliefs and attitudes that are coherent with religious values is expected among more religious individuals (Burn and Busso 2005; Mikołajczak and Pietrzak 2014, 2015). The predictive effect of *faith importance* for BS among our participants could be interpreted by the very nature of BS. Unlike HS, it implies subjectively positive attitudes towards women, and respect for and protection of women who embrace traditional values and roles, which corresponds with the teachings of the Catholic Church. It is reasonable to assume that individuals who consider religion to be more important accept religious values to a greater degree, and thus, strongly support BS; this then perpetuates traditional patterns of gender relations. Although higher education is associated with egalitarian attitudes (as suggested by Bartolac et al. 2011), the present study did not confirm its predictive effect. We may assume that individuals could change their existing attitudes, or form new ones, in relation to their greater egalitarianism during socialization. Additionally, this finding could be a negatively "false" as result of using only two educational response categories.

Finally, the finding of the low effect size of the chosen predictors make us believe that our participants shared some other common traits (e.g., values) not measured in this study. Furthermore, we may assume that individuals who endorse sexism to a greater extent are generally more inclined to join the military profession. However, the influence of military culture could be equally essential in the presence of sexism in the military. Therefore, the long-term exclusion of women from many combat duties and roles could have reinforced the stereotypical view of women as insufficiently competent for all military duties, and thus, bolstered the sexist beliefs of military personnel (Heinecken 2017; Young and Nauta 2013), especially among men.

*4.5. Types of Sexism Endorsement*

The clustering procedure identified three *sexism* types for both subsamples and revealed gender differences. Firstly, the findings showed that military women generally approved sexism to a lesser extent than their male counterparts. Gender differences were also visible in the structure of the sexist attitudes: *egalitarian*, *moderately egalitarian* and *traditional* types of women and *moderately egalitarian*, *traditional* and *hostile* types of men were revealed. *Traditional* women were the most numerous, and that type constituted slightly less than half of the female subsample. Compared to the members of other two types, they more strongly endorsed sexism (relatively low HS; moderately high BS_CGD and BS_HI). However, it is noticed that approximately one-quarter of all women already supported egalitarian values (*egalitarian* women), whereas the remaining women (one-quarter) were placed somewhere at the intersection of traditional and egalitarian values (*moderate egalitarian* women). Compared to *egalitarian* and *moderate egalitarian* women, *traditional* women were more likely to be younger, of lower *educational level* and *professional status*, more often came from rural areas, had served a shorter time in the military, and were more religious. These characteristics of *traditional* women are in line with the literature that associates lower education, growing up in rural areas and greater religiosity with traditional attitudes towards gender roles (Bartolac et al. 2011). We consider it particularly interesting that almost half of the women in our military sample (*traditional* women) rejected HS but still

endorsed BS, which could be considered a strong indicator of their exposure to sexism (Zakrisson et al. 2012).

When considering the male subsample, *traditional* men were also the most numerous and constituted approximately half of the subsample, whereas *moderate egalitarian* men constituted approximately one-third of the subsample. The third, *hostile* type consisted of men who strongly endorsed sexism (very high HS, very low BS_CGD, and high BS_HI). Although this represents the lowest number of men (approximately one-quarter), owing to the structure of their sexist beliefs, it deserves special consideration. It is justified to assume that *hostile* men are most at fault for maintaining an unequal working environment. *Hostile* men were more likely than *moderate egalitarian* and *traditional* men to be younger, with lower *education* and *professional status*, and a shorter military career. According to the structure of their attitudes, it could be assumed that *hostile* men are likely to adjust their behavior towards military women depending on the characteristics of the work situation or the women with whom they interact. In other words, they are likely to direct their hostility towards women who have deviated from traditional gender roles and opted for career advancement in the military, while displaying BS and "rewarding" women who have accepted positions and duties that comply with prescriptive gender norms. The finding of their very weak approval of benevolent beliefs, which idealize the stereotypical gender characteristics of women (BS_CGD), may also indicate more pronounced hostility towards women, and a preference for rivalry over cooperation with women. It is reasonable to assume that *hostile* men most strongly advocate and maintain the traditional pattern of gender relations in the military, so it is towards them that key policy measures should be directed. This prediction is based on the understanding that BS in society, and therefore in the military, is much more difficult to eliminate because it rests on the structure of personal relationships between women and men (Glick et al. 2000). Consequently, the effectiveness of policy measures in the military could be increased by their greater focus on reducing HS among men, and thus, both HS and BS among women could be reduced.

A moderately high appearance of sexism in the military work environment was confirmed. In such an environment, women could find it much harder to recognize and counter seemingly positive protective paternalism, flattery and affection from men (Lorenzi-Cioldi and Kulich 2015). Furthermore, with greater embracing of BS, women are much more willing to accept paternalistic justifications for their exclusion from certain activities and/or constraint of certain behaviors (Moya et al. 2007). Therefore, it is reasonable to assume that ostensibly protective restrictions are crucial obstacles to achieving gender equality in the military.

### 4.6. Limitations of Study

There are several considerable limitations to this study which restrict the possibility of generalizing the study findings. First, the use of a cross-sectional research design does not allow the determination of cause-and-effect relationships between the study measures. Further, a large portion of the variance in *sexism* measures remained unexplained, and other relevant individual, group and/or contextual variables could be used. Additionally, a greater effort could be undertaken to develop more refined measures of the used variables. As an illustration, in this study, a sufficiently representative number of responses was not used for some of the variables (e.g., the *educational level* variable), and some variables were measured with only a single item (e.g., the two measures of *religiosity*). Additionally, the results are based on self-reports of the study participants, and there is a chance that some of their responses were not valid. Finally, the broader cultural, working, historical and even religious specificity of the Croatian military environment, as well as the general Croatian society in which the study was conducted, must not be overlooked. Thus, the findings and conclusion derived from this study cannot simply be generalized and extrapolated to other cultural and/or work environments.

## 5. Conclusions

This study's validation of the ASI scale in a military sample confirmed that it is a satisfactory instrument for measuring sexism in the military. Given this study's findings, it is recommended to use BS subdimension measures instead of a general BS measure in future studies, where possible.

Among active-duty personnel, a moderately high endorsement of sexism, as well as significant gender differences, were identified: men endorsed HS and BS_HI to a higher extent than women, while women endorsed BS_CGD to a higher extent than men. Three types of women and men in the military were identified, differing in the endorsement of their sexist beliefs. Almost half of the female subsample were *traditional* women who—in comparison to *egalitarian* and *moderate egalitarian* women—were more likely to be younger, of a lower *educational level* and *professional status*, come from rural areas, have served a shorter time in the military, and be religious. Likewise, among men, almost half of the subsample comprised *traditional* men, with as much as one-quarter being *hostile* men (high HS, significantly low BS_CGD and moderate BS_HI). *Hostile* men were more likely than *moderate egalitarian* and *traditional* men to be younger, with a lower *education* and *professional status* and a shorter military career. These findings further support the more pronounced exposure of women to sexism in the military environment.

The findings of this study serve as an invaluable source of information that is important for better understanding gender relations in the military. Furthermore, the results suggest the need to raise awareness of the negative impact of gender prejudice on gender relations in the military work environment. Consequently, the military institution should provide clear guidelines for all commanders to emphasize their role in improving gender equality in the military. In the context of gender policy, we hope to encourage continued research interest in the systematic monitoring and development of targeted interventions that could contribute to policy improvement.

We think it is relevant to develop various psychosocial interventions capable of making military personnel more aware of the presence of sexism in the workplace, but also to educate them in embracing more egalitarian attitudes and living with them consistently. A moderately high approval of sexism in our sample suggests that education related to gender issues should be mandatory, particularly with regard to existing gender policy. Since gender prejudices are acquired through socialization, it is recommended that military personnel be provided with targeted education early in their military careers. This agrees well with the findings of this study, which showed that sexism is most pronounced among the younger and lowest-ranked service members. Moreover, it is advisable that all forms of military training undergo improvements to ensure the achievement of equal military competencies both for women and men. Consequently, women in the military might finally stop being dependent on the protection and paternalism of men. Ultimately, this could be regarded as an essential prerequisite for reducing and eliminating sexism and reaching gender equality. Finally, it could be interesting for future studies to explore the effectiveness of the proposed interventions in reducing the approval of gender prejudices in the military.

**Author Contributions:** Conceptualization, V.T.; methodology, V.T.; field research (investigation), V.T.; formal analysis, B.M. and V.T.; writing—original draft preparation, V.T., P.S. and B.M.; visualization, V.T. and B.M., writing—review & editing, V.T., P.S. and B.M. All authors have read and agreed to the published version of the manuscript.

**Funding:** This research received no external funding.

**Institutional Review Board Statement:** The study was conducted in accordance with the general ethical principles of conducting research with human participants, and approved by the Ethics Committee of the Faculty of Law, University of Zagreb, Republic of Croatia (9 October 2018).

**Informed Consent Statement:** All subjects gave their informed consent for inclusion before they participated in the study.

**Data Availability Statement:** Due to the data-collecting guidelines of the CAF, data are not available for sharing.

**Conflicts of Interest:** The authors declare no conflict of interest.

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
