# Peer review of "Initial Validation of the Ambivalent Sexism Inventory in a Military Setting"

_socsci, doi:10.3390/socsci11040176_

Round 1
Reviewer 1 Report
Dear editors,
thank you for offering me this great opportunity to be a reviewer of this manuscript named ‘’Initial validation of the ambivalent sexism inventory in the military setting". The manuscript explores the presence of sexism and gender prejudices in the military environment and validated a specific tool (ASI scale).
The manuscript is overall well written and easy to follow and results are interesting.
Based on my perception, the manuscript should be accepted after minor revisions. I have only some comments:
- Introduction could be shortened in order to make the reading more fluent
- Where the participants enrolled in light of some exclusion criteria? Please, specify.
- Did you find significant differences between men and women with regards to sociodemographic characteristics?
- With regards to English language, some typos are present. Minor spell check is required.
Author Response
Dear Reviewer,
Thank you for having given us one further opportunity to revise the manuscript Initial Validation of the Ambivalent Sexism Inventory in the Military Setting.
We have carefully considered the suggestions of the two Reviewers, and have implemented them as follows:
Point 1: Introduction could be shortened in order to make the reading more fluent
Thank you for your comments and suggestions. The introduction has been shortened. (lines 24- 121)
Point 2: Where the participants enrolled in light of some exclusion criteria? Please, specify.
Thank you. We believe that the answer is partly contained in section 2.1 Participants, and we will try to further clarify it. In addition, we point out that due to the data collecting guidelines of the CAF, some data are not presented in the article.
Only active military duty personnel were included in the study. The participants were selected from a list of all men and a list of all women in the CAF that were sorted alphabetically and by professional status, from the highest to the lowest rank. The proportion of women and men within each stratum was determined based on their actual numbers in each of the professional categories in the CAF. The systematic random sample interval was determined separately within each stratum, to ensure the desired size of approximately 500 participants in each stratum (every third woman and every ninth man). Thus, approximately equal sample sizes for men and women were guaranteed and statistical prerequisites for group comparison were met. The questionnaire was completed by 934 participants (out of a total 1007 selected). However, 39 participants were later excluded from the analyses due to missing values for more than 5% questions, and the final sample comprised 895 participants (445 men and 450 women).
Point 3: Did you find significant differences between men and women with regards to sociodemographic characteristics?
Thank you. We found some significant gender differences with regards to their sociodemographic characteristics. These results are shown in the table attached. However, our opinion is that these results are not important for this research. Namely, due to the identified gender differences, all statistical analyses were performed separately for each gender group.
Table xx. Differences of socio-demographic and professional characteristics of women and man
|
Variable |
Category |
Women |
Men |
Mann-Whitney U Test |
||||
|
Percent (%) |
Percent (%) |
U |
Z |
p= |
||||
|
Age |
20–29 years |
27.6 |
29.2 |
90776.0 |
1.20 |
0.23 |
||
|
30–39 years |
25.3 |
26.2 |
||||||
|
40–49 years |
27.8 |
30.6 |
||||||
|
50–59 years |
19.2 |
13.9 |
||||||
|
Educational level |
High school |
56.2 |
74.2 |
82167.5 |
-4.64 |
0.000 |
||
|
Bachelor's degree or higher |
43.8 |
25.8 |
||||||
|
Professional status |
Soldier (OR1–OR3) |
32.1 |
39.3 |
86712.0 |
3.36 |
0.001 |
||
|
NCO (OR4–OR9) |
32.6 |
38.9 |
||||||
|
Junior CO (OF1– OF2) |
23.2 |
12.1 |
||||||
|
Senior CO (OF3– OF5) |
12.1 |
9.7 |
||||||
|
Place of growing up
|
Village |
40.6 |
42.2 |
|
|
|
||
|
A very small town |
17.4 |
17.5 |
|
|||||
|
Smaller town |
17.6 |
24.9 |
94168.0 |
1.43 |
0.15 |
|||
|
City |
12.1 |
8.3 |
|
|
|
|||
|
Big city |
12.3 |
7.0 |
|
|
|
|||
|
Years of military service |
23 years or more |
36.2 |
34.0 |
94626.0 |
-0.11 |
0.91 |
||
|
18–22 years |
7.2 |
7.4 |
||||||
|
13–17 years |
7.0 |
7.0 |
||||||
|
8–12 years |
17.6 |
20.7 |
||||||
|
3–7 years |
11.1 |
13.5 |
||||||
|
0–2 years |
20.8 |
17.4 |
||||||
|
Faith importance
|
(completely irrelevant) 1 –2 |
4.7 |
9.1 |
91517.0 |
1.95 |
0.051 |
||
|
3–4 |
4.7 |
5.0 |
||||||
|
5–6 |
12.6 |
13.5 |
||||||
|
7–8 |
26.2 |
24.8 |
||||||
|
9–10 (very important) |
41.8 |
37.7 |
||||||
|
Religious practice |
1 (almost never) – 2 (rare) |
18.8 |
24.5 |
85758.5 |
3.61 |
0.000 |
||
|
3 (per year) – 4 (for holidays) |
25.9 |
34.4 |
||||||
|
5 (once a month) |
20.5 |
13.7 |
||||||
|
6 (per week) – 7 (more than weekly) |
34.8 |
27.4 |
||||||
Point 4: With regards to English language, some typos are present. Minor spell check is required.
Thank you for your suggestions. With regard to English language, we did a spell check of the whole text. Moreover, all manuscript changes were also highlighted into the text.
Kind regards

Reviewer 2 Report
The paper aims to present findings on a validation study of the Ambivalent Sexism Inventory (ASI) in the Croatian military population. In this respect, it sheds light on the changing gender relationships in the military sphere related to an increasing inclusiveness of women in it but still present strong gender stereotypes as well as the changing character of military professional activities that become more gender neutral. The desingna nd the methodology of the study are clearly described. The results are based on the application of e several multivariate statistical analyses such as factor analysis, correlation analysis, cluster analysis, etc. The findings are described in detail and linked to existing studies. Overall, they reveal the satisfactory metric properties of the ASI scale, the basic dimensions of ambivalent sexism and some specific results and effects of different variables in the application of the scale in the Croatian military sphere. The paper is very well written and my suggestion to author/s is to add a paragraph on the implications of the study with respect to sexism and gender inequality in a military context.
Author Response
Dear Reviewer,
Thank you for having given us one further opportunity to revise the manuscript Initial Validation of the Ambivalent Sexism Inventory in the Military Setting.
We have carefully considered your suggestions and have implemented them as follows:
Point 1: The paper is very well written and my suggestion to author/s is to add a paragraph on the implications of the study with respect to sexism and gender inequality in a military context.
We thank you for your comments. We revised the manuscript addressing your suggestions for improvement. In conclusion, we have made some changes in which we have more clearly highlighted the practical implications of this study (lines 752-775).
With regard to English language, we did a spell check of the whole text. Moreover, all manuscript changes were also highlighted into the text.
Kind regards
